# Roles of Estrogen, Estrogen Receptors, and Estrogen-Related Receptors in Skeletal Muscle: Regulation of Mitochondrial Function

**DOI:** 10.3390/ijms24031853

**Published:** 2023-01-17

**Authors:** Kenta Yoh, Kazuhiro Ikeda, Kuniko Horie, Satoshi Inoue

**Affiliations:** 1Division of Systems Medicine & Gene Therapy, Faculty of Medicine, Saitama Medical University, Hidaka, Saitama 350-1241, Japan; 2Department of Systems Aging Science and Medicine, Tokyo Metropolitan Institute of Geriatrics and Gerontology, Itabashi-ku, Tokyo 173-0015, Japan

**Keywords:** estrogen, estrogen receptor, estrogen-related receptor, muscle, metabolism, mitochondria

## Abstract

Estrogen is an essential sex steroid hormone that functions primarily in female reproductive system, as well as in a variety of tissues and organs with pleiotropic effects, such as in cardiovascular, nervous, immune, and musculoskeletal systems. Women with low estrogen, as exemplified by those in postmenopause, are therefore prone to suffer from various disorders, i.e., cardiovascular disease, dementia, metabolic syndrome, osteoporosis, sarcopenia, frailty, and so on. Estrogen regulates the expression of its target genes by binding to its cognate receptors, estrogen receptors (ERs) α and β. Notably, the estrogen-related receptors (ERRs) α, β, and γ are originally identified as orphan receptors that share substantial structural homology and common transcriptional targets with ERs. Accumulating evidence suggests that ERs and ERRs play crucial roles in skeletal muscles, such as muscle mass maintenance, muscle exercise physiology, and muscle regeneration. In this article, we review potential regulatory roles of ERs and ERRs in muscle physiology, particularly with regard to mitochondrial function and metabolism.

## 1. Introduction

Estrogen is a class of steroid hormones: it is primarily synthesized and secreted from mammalian ovaries, and regulates sexual organ differentiation and reproductive system development/maintenance [1,2,3,4,5]. Moreover, estrogen exhibits its various physiological functions in non-reproductive tissues and organs, including cardiovascular, nervous, immune, and musculoskeletal systems. In females, estrogen deficiency due to menopause can be related to the pathogenesis of disorders and diseases, such as atherosclerosis, dementia, hyperlipidemia, obesity, metabolic syndrome, type 2 diabetes, osteoporosis, sarcopenia, and frailty [6,7,8,9]. In this review, we focus on the contribution of estrogen and its cognate receptors as well as estrogen-related receptors to skeletal muscle functions.

## 2. Estrogen Actions in Muscle

Estrogen deficiency is clinically assumed to cause the onset of sarcopenia, which is a loss of skeletal muscle mass and strength [10,11,12,13], because a decrease in skeletal muscle mass and strength is usually greater in menopausal women than age-matched men [14]. In general, skeletal muscle mass directly correlates with muscle function, thus the prevention of muscle mass reduction or muscle atrophy will improve the quality of life for aged women [12,15]. Because muscle strength is often more severely impaired than muscle mass in menopause-related sarcopenia, the preservation of muscle quality is also important [11,12,16]. Several studies show that estrogen hormone replacement therapy (HRT) prevents the reduction in skeletal muscle mass and strength in menopausal women [9,17,18,19]. For example, a meta-analysis in postmenopausal women indicates that the estrogen-based HRT provides a beneficial effect on muscle strength [20]. Recently, another meta-analysis showed that the beneficial effect of estrogen-based HRT history on skeletal muscle size and quality lasts even after the therapy cessation [19]. Importantly, many factors, including estrogen dosage, age, and menopause, may modulate the efficacy of estrogen-based HRT [21]. While the evidence remains to be defined, it is likely that estrogen facilitates the prevention of menopause-related loss of muscle mass and strength [17].

Estrogen may also impact training and physical performance in female athletes. Fluctuating estrogen levels during menstrual cycle and the decline in estrogen levels associated with athletic amenorrhea can modulate physical performance, although further large-scale studies are required to define the effects of estrogen on exercise performance [22,23].

Estrogen-deficient animals generated by ovariectomy (OVX) can be useful for a model of human menopause to examine the effects of estrogen supplementation on muscle mass and strength [24]. Because HRT is usually performed as combined hormone therapies involving estrogen and other hormones, such as progesterone, it is rather difficult to evaluate distinct functions of individual hormones. OVX-treated animal models followed by estrogen supplementation will help us to dissect the estrogen action on muscle functions, although few such studies were conducted.

In a study of ovarian-senescent mice generated by 4-vinylcyclohexene diepoxide (VCD) treatment, estrogen treatment enhanced soleus muscle strength and reduced fatigue, whereas it did not alter muscle size [25] (Table 1). In our study, OVX mice followed by estrogen supplementation ran for a longer time than OVX mice without supplementation in a treadmill endurance test, implicating that muscle endurance capacity is enhanced by estrogen [26]. Among those mice groups, the weights of skeletal muscles were not altered, suggesting that the estrogen-dependent alteration of exercise endurance is likely due to muscle quality modification rather than due to muscle mass alteration. Transcriptomic analysis in slow-twitch fiber-dominant soleus muscles identified uncoupling protein 3 (Ucp3) as an estrogen-repressed gene. Because UCP3 contributes to proton leakage and diminishes energy production in mitochondria, estrogen may efficiently produce more energy in muscles by repressing mitochondrial uncoupling. Another report showed that OVX mice exhibit decreases in the muscle force generation in a grip test [27] and in the cross-sectional area of the tibialis anterior (TA) muscle, which is rescued by estrogen treatment. The report further indicated that OVX mice possess the decreased cross-sectional area of fast type fibers in TA muscle. These observations suggest that estrogen may inhibit muscle atrophy by shifting the fiber type composition toward faster type in female mice. Moreover, the report implicates that OVX mice show impaired muscle regeneration due to the decreases in expansion, differentiation, and self-renewal of muscle satellite cells. Thus, estrogen is assumed to be involved in the maintenance of muscle function in mice by supporting muscle strength and facilitating muscle regeneration. Supportively, Chaiyasing et al. reported that estrogen administration increased the diameter of regenerated myotubes in OVX mice after cardiotoxin-induced injury, suggesting that estrogen is essential for the myoregeneration process [28].

In the C57BL/6J mouse strain, females possess larger and smaller amounts of type I and type II myosin heavy chains, respectively, in the soleus muscle fibers compared with males, showing greater exercise endurance than males [29]. It is notable that OVX abrogates the enhanced exercise endurance in female mice, and contrarily, estrogen administration increases exercise capacity in males to the similar level of intact females. Nitric oxide synthase, a known downstream target of estrogen, is a candidate that mediates the enhanced exercise performance in females. A recent transcriptomic study demonstrated that pyruvate dehydrogenase kinase isoform 4 (Pdk4), a key mitochondrial enzyme to inhibit pyruvate dehydrogenase in glycolysis, is upregulated by estrogen treatment in ovariectomized mice [30]. The report further revealed that the fatty acid-dependent oxidative metabolism is enhanced by estrogen treatment. Since PDK4 contributes to a metabolic shift from glucose to fatty acids as major energy source, it is implicated that the estrogen-induced Pdk4 activation allows predominant fatty acid β-oxidation in female, which supports the notion that skeletal muscles preferentially utilise fatty acid β-oxidation for energy production in females [31].

## 3. Types and Structure of Estrogen Receptors

Estrogen receptors (ERs) are “nuclear steroid receptors”, which belong to the NR3 class of nuclear receptor superfamily [32]. There are two nuclear ERs, ERα (NR3A1) and ERβ (NR3A2), which share structural characteristics, whereas tissue distributions and abundance are different between them [33]. In humans, ERα and ERβ are encoded by *ESR1* and *ESR2* genes, respectively. Estrogen regulates the target cells through binding to the ERs [34]. ERs have a structure consisting of the N-terminal domain (NTD), DNA-binding domain (DBD), hinge region, and ligand-binding domain (LBD). The NTD is the largest domain that has transcriptional activation function (AF-1), exhibiting a constitutive transcription activation ability regardless of ligand binding [35,36]. NTD has an intrinsically disordered region that lends the ability of allosterically controlling ERs with ligands, DNA, and other transcriptional modulators. The DBDs contain two zinc finger structures and share 97% of the amino acid identity between ERα and ERβ. Amino acids at the base, which is proximal to amino terminal, is called proximal box, or, P box. P box is critical for recognizing the specific DNA sequence of the estrogen response element (ERE) motif (GGTCAnnnTGACC) [37]. On the other hand, amino acids at the base of the second zinc finger are termed distal box, or, D box, which contributes to ER dimer formation. The hinge region is important for nuclear localization of the ER protein. Moreover, the hinge region contains part of the C-terminal extension of the DBD and the intrinsically disordered region of the sequence offers the flexibility of allosteric regulation of ER interactions similar to the NTD [36]. The LBD consists of 12 helix structures, folding into a complex pocket with a globular structure. The 11α helices form the ligand-binding pocket, which contains the site of high-affinity interaction with its ligand. Binding to the estrogen ligand repositions helix 12, which is part of the second transcriptional activation function 2 (AF-2) [38]. When estrogen binds with ERs, estrogen forms hydrogen bonds within the ligand binding pocket in the LBD of both ERα and ERβ, resulting in activation of AF-2. AF-2 exhibits estrogen-dependent transcriptional activity and interacts with mediators of chromatin accessibility and RNA transcription rate [36]. The structural position of helix 12 critically determines the transcriptional activity which exhibits the ligand-dependent activation function AF-2 because it generates an interaction surface with the coactivators [39]. Without ligand binding, AF-2 suppresses the activity of AF-1 [40]. When binding to the ligand, the structure of ERs changes significantly. ERα can also activate transcription through the association with other DNA-bound transcription factors without DNA binding [41]. ERα also interacts indirectly to nuclear DNA by its direct (protein–protein) interaction with other DNA-bound transcription factors, e.g., SP1, AP-1, and RUNX1 [41]. This mechanism of ERα transcriptional activation is called tethering (Figure 1A) [42].

When the ERs bind with ligand, the ligand-bound ERs can bind to EREs in the genome and stimulate structural changes of chromatin to recruit coactivators and factors involved in the RNA polymerase transcription machinery [42].

Apart from the roles of nuclear ERs in terms of their transcriptional regulation, estrogens are also reported to rapidly activate signaling molecules such as cyclic AMP, inositol triphosphate, and activate the protein kinase (i.e., mitogen-activated protein kinases) pathway through a membrane-type G protein-coupled ER (GPER), or GPR30, which is predominantly localized to the intracellular membranes (i.e., membranes of the endoplasmic reticulum and Golgi apparatus) [43,44,45,46]. The role of GPER in muscle exercise remains elusive.

## 4. Estrogen-ER Signaling in the Regulation of Skeletal Muscle Function and Mitochondria

Estrogen is considered to exhibit beneficial effects on mitochondrial function and clinical outcomes, including improved metabolism in type 2 diabetes and sarcopenia prevention. Thus, the function and mechanism of estrogen was considerably investigated with focus on disease prevention [10,47,48,49]. Intriguingly, skeletal muscle in estrogen-treated female rats exhibits increases in mitochondrial mass, anti-oxidant protection, and oxidative phosphorylation compared with male rats [50,51]. On the contrary, skeletal muscle in OVX female animals exhibits a decrease in oxygen consumption in addition to the reduced expression of mitochondrial biogenesis-related genes and mitochondrial remodeling factors associated with enhanced hydrogen peroxide generation [26,52]. Torres et al. also show that estrogen deficiency in mice decreases the activity of mitochondrial respiratory complex I in muscle, which is recovered by estrogen treatment [53].

Skeletal muscle is a crucial organ in whole-body energy metabolism, particularly in oxidative metabolism and insulin-stimulated glucose uptake [54,55,56]. Estrogen is implicated in the skeletal muscle metabolism because muscle ERα mRNA (*ESR1*) level was decreased in women who suffered from metabolic syndrome and it was inversely associated with adipose tissue mass and fasting insulin level [57].

In muscle-specific ERα knockout (MERKO) mice, single muscle fibers of the animals fatigued faster than those of control mice (Table 2). Moreover, MERKO mice show the impaired glucose tolerance and insulin resistance, and the lipid accumulation in skeletal muscle, which is paralleled by enhanced inflammation signal, such as the phosphorylation of c-Jun N-terminal kinase 1/2 (JNK 1/2) and IκB kinase β (IKKβ) [57,58]. Experiments using shRNA-mediated *Esr1*-knockdown muscle cells implicate the impaired fatty acid oxidation, which may explain the lipid accumulation in MERKO mice muscle [57]. MERKO mice also show a reduction in basal and stimulated oxygen consumption rates, a decrease in mtDNA replication rate, and a gain of reactive oxygen species (ROS) production, indicating mitochondrial dysfunction in muscle. Analysis of MERKO mice and C2C12 myoblastic cells with *Esr1* silencing revealed that the decreased expression of mtDNA polymerase Polg1 possibly results in defects in the muscle mitochondrial dysfunction [57]. Moreover, MERKO mice exhibit the alteration of mitochondrial morphology, i.e., the elongated and highly interconnected mitochondria, by strong suppression of mitochondrial fission signaling. In fact, ERα deficiency increases inhibitory phosphorylation of DRP1 protein, a key factor of the mitochondrial fission process. As mentioned above, MERKO mice exhibit the phenotype of dysfunctional mitochondria in muscle, thus ERα will function to maintain mitochondrial function and health as a guardian against mitochondria-related disorders in women.

Another line of skeletal muscle-specific ERα-deficient mice (skmERαKO) was generated and showed low strength/contractility in several skeletal muscles [59,60]. In skmERαKO mice, the extensor digitorum longus muscles exhibit impaired eccentric and submaximal/maximal isometric force, and the soleus muscles exhibit easy fatigue with diminished force recovery. In addition, in vivo maximal torque/power generation decreased in plantarflexors and dorsiflexors in skmERαKO mice, along with low phosphorylation of myosin regulatory light chain. These findings indicate that ERα mediates beneficial effects of estradiol on muscle strength.

In the recent report, Collins et al. show that estrogen maintains the satellite cell number in the muscle of female mice and human [63]. In addition, muscle stem cell-specific ERα knockout mice demonstrated that ERα is indispensable for satellite cell maintenance, self-renewal, and prevention from apoptosis, which is required for muscle regeneration. The study suggests that estrogen-ER signaling axis may influence muscle health by modulating muscle cell physiology, as well as by affecting the regulatory system in muscle stem cells. In addition, conditional knockout mice of ERα in skeletal muscle show osteopenia, suggesting that estrogen signaling regulates expression of myokines that regulate the differentiation and activity of osteoclasts [64].

In muscle-specific ERβ-knockout (mERβKO) [61], there was no difference in treadmill exercise performance between mERβKO and control mice; however, the grip strength test revealed the impaired absolute mean maximum strength in female mERβKO mice. In addition, fast-type dominant muscle mass decreases in young female mERβKO mice. Moreover, muscle-specific ERβ-knockout and satellite cell-specific ERβ-knockout mice models show that muscle-specific ERβ deficiency causes a decrease in muscle mass and strength in female mice. Inactivation of ERβ also decreases the proliferative capacity of satellite cells [61].

Our group generated Mck-caERα transgenic mice, which express a constitutively active mutant of human ERα (caERα) with a Y537S substitution in muscle [62,65,66]. The Mck-caERα transgenic mice muscle-specifically express caERα that exhibits ligand-independent transcriptional activation [62]. The Mck-caERα transgenic mice have no changed weights of skeletal muscles normalized to body weight, but prolonged running time in a treadmill exercise test, indicating that ERα activation in muscle elevated endurance capacity in mice. Transcriptomic analysis in quadriceps femoris muscles revealed that the following pathways are enriched in upregulated genes: pathways related to (1) fatty acid metabolism such as acetyl-CoA carboxylase alpha (*Acaca*), fatty acid synthase (*Fasn*), elongation of long chain fatty acids family 6 (*Elovl6*), and stearoyl-Coenzyme A desaturase 1 (*Scd1*); (2) insulin sensitivity such as adiponectin (*Adipoq*) and peroxisome proliferator-activated receptor gamma (*Pparg*); and (3) growth factors such as insulin-like growth factor 1 and 2 (*Igf1* and *Igf2*). These findings suggest that estrogen signaling potentially activates exercise endurance in skeletal muscle via the modulation of metabolism-related gene expression.

Our group also reported that caERα overexpression in myoblastically differentiated C2C12 cells downregulates mitochondrial uncoupling protein 3 (*Ucp3*) expression and upregulates intracellular ATP levels [26]. Thus, estrogen may contribute to efficient ATP generation in mitochondria by blocking energy dissipation. Subsequently, nuclear receptor subfamily 4 group A member 1 (*Nr4a1*) was also demonstrated as an estrogen responsive gene in the caERα-overexpressed C2C12 cells [67]. Since NR4A1 stimulates the respiration of skeletal muscle mitochondria [68], NR4A1 may also mediate estrogen function in muscle.

It is assumed that estrogen enhances oxidative metabolism in muscle and affects mitochondrial function contributing to disease prevention. Therefore, the investigation was progressed for precise actions of estrogen in mitochondria, which are the important cellular organelle responsible for controlling oxidative metabolism. Mitochondria contain their own DNA of the mitochondrial genome in the matrix. Mitochondrial genome is inherited maternally and exists as a circular, double-stranded DNA composed of 16,569 base pairs in humans [69]. As the mitochondrial genome encodes a small number of mitochondrial genes, including transfer RNAs, mitochondrial ribosomal RNAs, and protein subunits of the electron transport chain complexes, many mitochondrial genes are encoded in the nucleus. Thus, the coordination of the transcriptional events between the mitochondria and nucleus is required to maintain metabolic homeostasis [70].

Our research group demonstrated that cytochrome c oxidase (COX) subunit 7a-related protein (COX7RP), an estrogen-responsive gene that we originally identified from breast cancer cells, modulates mitochondrial metabolism in vivo [71]. Notably, the *Cox7rp*-knockout mice (*Cox7rp*KO) exhibited shorter running distance and time in a treadmill exercise test, indicating less endurance. In addition, *Cox7rp*KO mice possess decreased mitochondrial respiratory complex activities and less formation of mitochondrial respiratory supercomplexes in skeletal muscle. In contrast, COX7RP-transgenic mice (*COX7RP*-Tg) have enhanced exercise endurance and increased cytochrome *c* oxidase (complex IV) activity. Our research also showed that mitochondrial ATP synthesis was decreased in *Cox7rp*KO mice. Collectively, these findings suggest that COX7RP mediates estrogen-dependent activation of mitochondrial respiration, which may play a critical role in energy metabolism [71,72].

## 5. Similarities of Estrogen-Related Receptors with ERs

Estrogen-related receptors (ERRs), ERRα, ERRβ, and ERRγ, were identified as nuclear receptors with substantial sequence similarities to ERα. Both ERs and ERRs belong to the NR3 subgroup of the nuclear receptor superfamily [32]. Despite sharing sequence homologies with ERs, ERRs do not bind with any endogenous ligand including estrogen, but require the binding with a transcriptional coactivator or a protein ligand to exert transcriptional activity [73].

All ERRs have the structural features of the steroid nuclear receptor superfamily, including a non-conserved amino terminal domain (NTD), a central DNA-binding domain (DBD), and a ligand-binding domain (LBD) providing a docking surface for coregulators [73]. The NTD of nuclear receptors is a non-conserved and unstructured region containing AF-1, and is often modified post-translationally. Because ERRs have substantial amino acid identity in their NTDs, the domain may play an important role in transcriptional activities [73,74]. The two highly conserved zinc finger motifs in the DBD of ERRs bind to a consensus DNA sequence (TCAAGGTCA) called the estrogen-related response element (ERRE) (Figure 1B). ERRs bind to ERRE as a monomer or a homodimer [75]. The LBD of the ERRs contains a well-conserved helix motif with AF-2. Interestingly, ERRs can bind to ERE and, conversely, ERα can bind to ERRE, suggesting the overlapping regulatory networks for ERα and ERRs [73,76]. For example, the osteopontin gene promoter is stimulated through ERRE sequences by ERRα, as well as by ERα [77]. Moreover, it is noted that estrogen stimulates the expression of the ERRα gene in mouse heart and uterus. ERα binds with multiple steroid hormone response element half-sites (MHREs) that are conserved in the ERRα gene promoter. Further analysis of the crosstalk between ER and ERR nuclear receptor subgroups will reveal their common targets and precise mechanism in target cells and tissues.

## 6. ERRs Signaling in the Regulation of Skeletal Muscle Function and Mitochondria

It was suggested that ERRs are initially related to mitochondrial function [78], thus their function in muscle was analyzed earlier than that of ER (Table 3). Muscle-specific ERRγ transgenic mice have markedly redder muscles and all hind limb muscles possess the same color as the soleus muscle [79]. The muscle-specific ERRγ transgenic mice exhibit decreases in muscle weight of glycolytic and mixed fiber muscles, and show an increase in numbers of large mitochondria. Notably, in a treadmill endurance test, the muscle-specific ERRγ transgenic mice exhibited significant increases in the running distance and the peak oxidative capacity compared with the wild-type (WT) controls. During endurance exercise, the respiratory exchange ratio, which is the ratio of the metabolic production of CO_2_ to the uptake of O_2_, is smaller in the transgenic mice than control mice, implicating an energy substrate preference from glucose oxidation to fatty acid oxidation. Muscle mitochondrial activity such as succinate dehydrogenase and aconitase enzyme activities strongly increases in the ERRγ transgenic mice. Gene expression pattern indicates the conversion of muscle fiber types from fast type II toward slow type I muscle fibers. On the contrary, the ERRγ heterozygotes knockout mice exhibit a shorter running distance in a treadmill endurance exercise test. Fatty acid uptake and oxidation genes were decreased in the muscle of the ERRγ heterozygotes knockout mice, indicating a decrease in fatty acid utilization as energy source.

Another report of muscle-specific ERRγ transgenic mice showed that ERRγ upregulates the expression of oxidative metabolism genes (*Ucp3*, *Pdk4*, *Cycs*, *Cox5a*, and *Lpl*) and oxidative myofibers [myosin heavy chain 1a (*Mhc1a*) and *Mhc2a*] in skeletal muscle [80]. In addition, angiogenesis and muscle vascularization were induced by ERRγ, suggesting fatigue resistance in slow-twitch myofibers. Transcriptomic analysis revealed the increased expression of angiogenic genes, such as vascular endothelial growth factor A (*Vegfa*). The ERRγ-overexpressing (ERRGO) transgenic mice have an elevated oxygen consumption rate along with the increased oxidative metabolism and blood supply to skeletal muscles. The ERRGO mice have a lower respiratory exchange ratio (RER) compared to the WT mice, suggesting a tendency to prefer oxidation of fatty acids over carbohydrates. ERRγ transgenic mice exhibit a longer running time than WT mice in a treadmill endurance test. ERRGO mice gained less weight than WT controls on a high-fat diet. It is therefore assumed that ERRγ upregulates oxidative metabolism and blood supply to skeletal muscle, leading to increased oxygen consumption, better exercise endurance, and resistance to weight gain.

Skeletal muscle-specific double knockout mice of the *Esrrg* and *Esrrb* genes (ERRβ/γ dmKO mice) were generated to assess distinct effects of PPARβ/δ and PPARα on muscle fiber-type determination [81]. Gene expression analysis revealed that slow-twitch fiber-related gene expression and type I myosin heavy chain-positive fibers were decreased in ERRβ/γ dmKO gastrocnemius muscle. In a treadmill endurance test, the ERRβ/γ dmKO mice exhibited a shorter running distance compared with the WT controls, which is consistent with the changes in muscle fiber-type composition. In human muscle biopsies, a strong positive correlation was found between *ESRRG* expression levels and the determinants of muscle endurance (i.e., type I fiber percentage, ATPmax, and VO_2_max). Furthermore, miRNA-499 expression is also correlated with the determinants of muscle endurance. Taken together, it is assumed that the ERRγ/miRNA-499 axis functions under the control of PPARβ/δ to regulate the type I muscle fiber characteristics.

ERRα knockout mice show significant decreases in heart, gastrocnemius, soleus, and quadricep muscle mass compared with WT mice [82]. The ERRα knockout mice have a shorter running time than WT mice in a forced treadmill exercise challenge, indicating reduced exercise tolerance. Metabolic analysis shows that the ERRα KO mice have a higher respiratory exchange ratio (RER), suggesting that the mice are more dependent on carbohydrates than lipids as substrates for an energy source. In addition, ERRα target genes, including *Cycs*, *Idh3g*, and *Pdha1*, all of which are important for mitochondrial energy metabolism, are downregulated at both pre- and post-exercise time points in the skeletal muscle of ERRα KO mice. In the ERRα KO skeletal muscle, tricarboxylic acid cycle (TCA) cycle intermediates, such as citrate, cis-aconitate, and α-ketoglutarate, are accumulated after exercise, suggesting a compensatory or adaptive response against the impaired mitochondrial oxidative capacity. On the other hand, the amounts of TCA cycle intermediates, such as succinate and malate, decreased in the ERRα KO mice skeletal muscle after exercise via downregulation of α-ketoglutarate dehydrogenase, which is responsible for the conversion of α-ketoglutarate to succinyl coenzyme A. Taken together, these findings imply that substantial alterations occurred in the ERRα KO mice in terms of the availability, transformation, or replenishment of several metabolic substrates critical for energy production.

ERRs also play a critical role in muscle mitochondrial biogenesis. ERRs are involved in the activation of many aspects of mitochondrial energy metabolism, including fatty acid oxidation, the TCA cycle, and oxidative phosphorylation (OXPHOS) [78]. Villena et al. reported that ERRα is essential for adaptive thermogenesis through directly activating genes important for mitochondrial function [83]. ERRα also regulates muscle repair and regeneration [84,85]. Moreover, ERRγ stimulates oxidative metabolism and mitochondrial biogenesis, and increases slow type I fibers in muscle [80,86]. Combined with the notion that the gene expression profile activated by ERRγ is similar to that of the red oxidative fiber-type muscle and ERRγ also regulates the expression of muscle-related miRNA, ERRγ-regulated pathways provide a mechanism in which ERRγ coordinately control slow type I myosin heavy chain expression and high oxidative metabolism [79,81]. A schema for ERs and ERRs functions in muscle cells and mitochondrial metabolism is shown in Figure 2.

In a study of high-fat diet-induced obese mice performed by Sopariwala et al., muscle-specific ERRα overexpression induced the expression of angiogenic factors and promoted neo-angiogenesis in skeletal muscle, muscle ischemic revascularization, and post-ischemic muscle recovery [85]. Transcriptomic analysis by RNA-sequencing revealed that the angiogenic gene pathway is most prominently activated by ERRα overexpression in the skeletal muscle, which suggests that ERRα may affect skeletal muscle physiology by mitochondrial regulation and may relate to ischemic revascularization and muscle recovery, at least in obesity [85]. In addition to skeletal muscle physiology regulation, ERRα and ERRγ play critical roles in cardiac myocyte maturation, acting as transcriptional activators of adult cardiac metabolic and structural genes [87,88]. Most recently, ERRγ was demonstrated as a hallmark of oxidative stress induced by mitochondrial disruption [89]. Intracellular ROS generation causes the diminution of ERRα and accumulation of ERRγ, and ERRγ inhibition increases an inhibitory effect of paclitaxel on breast cancer cell growth.

While ERRs are naturally orphan receptors without endogenous ligands, they require the transcriptional coregulators for their transcription activation. Indeed, the ERRs are tightly regulated by peroxisome proliferator-activated receptor (PPAR) coactivator 1 α (PGC-1α) and PGC-1β [90]. Thus, PGC-1s and ERRs collaboratively modulate several aspects of mitochondrial function [90,91]. For example, exercise enhances mitochondrial turnover in muscle, which is accomplished by mitophagy and the renewal of mitochondria through mitochondrial biogenesis [92]. In PGC-1α KO mice, mitophagy and mitochondria renewal are impaired in response to exercise, indicating that PGC-1α coordinates mitophagy with mitochondrial biogenesis [93,94]. In mice, PGC-1α and ERRα expression gradually decline with aging in skeletal muscle, correlating with the alteration of mitochondrial function and tissue integrity [95]. Moreover, muscle-specific knockout and overexpression of PGC-1α in old mice accelerate and mitigate the aging-related impairment of muscle function, respectively. Notably, the muscle-specific PGC-1α transgenic mice exhibit improved endurance exercise in the advanced aged animals [91]. The PGC-1/ERR axis will act as a major molecular circuitry that orchestrates mitochondrial function and metabolism in muscle [82].

## 7. Potential Clinical Implications of ER- and ERR-Dependent Muscle Disease Models and Tissue Engineering

Current knowledge of ERs and ERRs on skeletal muscle will potentially provide a clue to the development of alternative therapeutic options, particularly in the context of tissue engineering. Studies using satellite cells isolated from OVX mice and satellite cell-specific ERβ knockout mice show that the deficiency of estrogen signaling decreases the self-renewal and differentiation abilities in satellite cells [27,61]. In rats, estrogen increases the number of satellite cells after exercise [96,97] and stimulates muscle regeneration after injury in which ERβ plays a primary role [98]. These findings imply that estrogen enhances the proliferation of muscle stem cells [99]. Interestingly, it is reported that estrogen can improve the production of functional myogenic differentiated cells from adipose-derived stem cells, which are used with a nano-scaffold to treat stress urinary incontinence [100]. In C2C12 cells, ERRα overexpression stimulates the myoblastic differentiation, while ERRα inverse agonist XCT790 impairs myotube formation with fewer mitochondria and reduces sarcomeric assembly. These findings indicate that ERRα plays an important role in skeletal myocyte differentiation [101]. In addition, an ERRγ agonist enhances cardiac maturation with transverse tubule formation in cardiomyocytes derived from human-induced pluripotent stem cells (hiPSCs) [102]. Thus, ERRγ signaling is also useful for muscle disease modeling and regenerative medicine. Further investigation will clarify the mechanisms of ERs and ERRs in muscle stem cells, which are critical for both fundamental tissue repair and regeneration.

## 8. Conclusions and Perspectives

The life expectancy of women substantially increased over the last century in both developed and developing countries, indicating that postmenopausal women live much longer than before. Various muscle disorders including loss of muscle mass and strength, sarcopenia, and frailty are well observed in postmenopausal women, thus they are potentially related to the decline in endogenous estrogen secretion from ovary. Considering that the elderly women contribute a lot to the society and household, their muscle disorders are critical social issues from both medical and economic viewpoints. For the maintenance of physical activity necessary for healthy life, ERs and ERRs are essential factors that control the complex regulatory network of metabolism and mitochondrial function in skeletal muscle. The elucidation of gene regulatory pathways exerted by ERs and ERRs in muscle is expected to be applied to alternative diagnostic and therapeutic options for muscle-related diseases, particularly for aged women with estrogen deficiency. New insights and further understanding of ERs and ERRs in the metabolism and mitochondrial function in muscle will improve quality of life by preventing muscle loss and frailty.

## Figures and Tables

**Figure 1 ijms-24-01853-f001:**
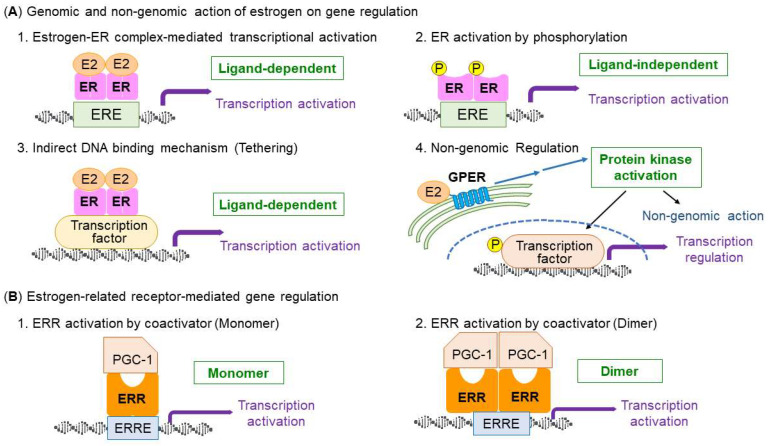
Classical nuclear estrogen receptors (ERs) and membrane-type G protein-coupled estrogen receptor (GPER). (**A**) Endogenous estrogens, including 17β-estradiol (E2), bind to two types of nuclear ERs (i.e., ERα, ERβ) or membrane-type GPER, which is predominately localized to the intracellular membranes. Estrogen activates nuclear ERs, inducing the dimerization of the receptors and binding of receptors to estrogen responsive elements (EREs) in the genome. Alternatively, ERs bind with other classes of transcription factors through protein–protein interactions and the complexes bind to the genome. GPER stimulates the activation of protein kinases (i.e., MAPK and PI3K/Akt), which mediates nongenomic actions of estrogen and contributes to transcription factor regulation [43]. (**B**) Schematic model for transcriptional regulation by ERRs and its coactivator PGC-1. ERRs preferentially bind to estrogen-related receptor responsive elements (ERREs) in the genome as a monomer or a dimer. PGC-1 is a necessary coactivator for the activation of ERR transcription activity.

**Figure 2 ijms-24-01853-f002:**
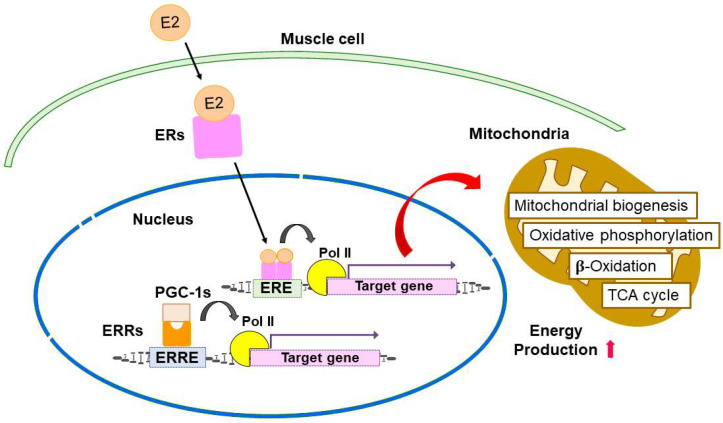
ERs and ERRs regulate muscle functions and mitochondrial metabolism.

**Table 1 ijms-24-01853-t001:** Phenotypes of ovariectomized mice on muscle performance.

Mouse	Experimental Condition	Phenotype	Other Phenotypes in Muscle	Reference
Ovarian-senesce by chemical, 4-vinylcyclohexene diepoxide (VCD) treatment followed by estrogen treatment for 8 weeks	In vitro muscle contractility test	Estrogen replacement increases muscle strength compared with no estrogen treatment mice.	No difference in soleus muscle size	[25]
OVX followed by estrogen administration for 10 weeks	Treadmill endurance test	Endurance is increased by estrogen administration.	Mitochondrial uncoupling protein 3 (UCP3) is upregulated by ovariectomy and downregulated by estrogen administration.	[26]
OVX for 24 weeks	Grip strength test	Grip force is decreased by OVX.	Increase in the proportion of fast twitch type fibers in the tibialis anterior muscle. This fiber-type shift was recovered by estradiol. Satellite cells were impaired in OVX mice.	[27]
OVX followed by estrogen administration for 2 weeks	Treadmill endurance test	Endurance is increased by estrogen administration.	Nitric oxide synthase activity is increased in females compared with males.	[29]

**Table 2 ijms-24-01853-t002:** Phenotypes of transgenic or knockout mice for ERs on muscle performance.

Mouse *	Experimental Condition	Phenotype	Other Phenotypes in Muscle	Reference
Muscle-specific knockout of ERα (MERKO)	In vitro muscular force and endurance test	Single muscle fibers from MERKO mice fatigued faster than fibers from control muscle.	Reduced oxygen consumption rates, excessive production of reactive oxygen species in mitochondria, and morphological abnormalities of mitochondria, indicating an impairment of fission-fusion dynamics. Reduction in mitophagy.	[57]
Muscle-specific knockout of ERα (skmERαKO)	In vitro muscle contractile test	Greater fatigability and impaired recovery from fatigue in muscles from skmERαKO mice	Phosphorylation of myosin regulatory light chain (RLC) was decreased in muscles from skmERαKO compared with WT mice.	[59]
Muscle specific estrogen receptor α knockout mice (skmERαKO)	Ex vivo or in vivo testing of muscle contractility	Smaller force and fatigability of soleus muscles. Less torque in in vivo plantar flexor muscle contractility.		[60]
Muscle-specific ERβ-knockout (mERβKO)	Grip strength test	The absolute mean maximum strength was slightly decreased only in female KO mice compared with control mice.	Fast-type dominant muscle mass decreases in young female KO mice. There was no difference in running performance.	[61]
Muscle-specific constitutively active ERα transgenic (Mck-caERα)	Treadmill endurance test	Increased endurance	Genes related to lipid metabolism, insulin signaling, and growth factor signaling were upregulated.	[62]

* Name of mice used in the literature are shown in brackets.

**Table 3 ijms-24-01853-t003:** Phenotypes of transgenic or knockout mice for ERRs on muscle performance.

Mouse *	Experimental Condition	Phenotype	Other Phenotypes in Muscle	Reference
Muscle-specific ERRγ and VP16ERRγ transgenic [ERRγ (N-TG) and VP16ERRγ (TG)]	Treadmill endurance test	Increased endurance	Decrease in muscle weight of glycolytic and mixed fiber muscles, increase in numbers of large mitochondria, and improved oxidative capacity and mitochondrial enzymatic function.	[79]
Heterozygotes knockout of ERRγ (HET)	Treadmill endurance test	Decreased endurance	Impaired mitochondrial oxidative metabolism.	[79]
Muscle-specific ERRγ transgenic (ERRGO)	Treadmill endurance test	Increased endurance	Increase in mitochondrial respiration, type I fiber specification, and vascularization.	[80]
Muscle–specific ERRγ/ERRβdouble knockout (ERRβ/γ dmKO)	Treadmill endurance test	Decreased endurance	miRNAs (miR-499 and miR-208b) and type I fiber-related genes were reduced, suggesting that ERRs are required for type I fiber formation.	[81]
ERRα knockout (ERRα KO)	Treadmill endurance test	Decreased endurance	Decreased muscle mass and reduced expression of many genes involved in mitochondrial oxidative metabolism	[82]

* Name of mice used in the literature are shown in brackets.

## Data Availability

Not applicable.

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
