# Peer review of "Roles of Estrogen, Estrogen Receptors, and Estrogen-Related Receptors in Skeletal Muscle: Regulation of Mitochondrial Function"

_ijms, 2023, doi:10.3390/ijms24031853_

Round 1

Reviewer 1 Report

This is an interesting review with sufficien references and a novel summery of advancements on the field.

Minors:

l 44- “prevention of muscle quality”- perhaps preservation of… instead?

Table 1: last ref. item: “NO synthase activity increased in females than males” should be corrected

1 183: what is ESR1? please align with others

l 195:  Esr1 knock down mice please align nomenclature

Author Response

Response to reviewers’ comments

Reviewer 1

This is an interesting review with sufficient references and a novel summery of advancements on the field.

Ans: We thank this reviewer for her/his valuable comments. As suggested by this reviewer, we respond to every comment point-by-point.

Minors:

l 44- “prevention of muscle quality”- perhaps preservation of… instead?

Ans: As pointed out by the Reviewer, “prevention” was corrected to “preservation” (line 43, page 1).

Table 1: last ref. item: “NO synthase activity increased in females than males” should be corrected

Ans: As pointed out by the Reviewer, the sentence was corrected to “Nitric oxide synthase activity is increased in females compared with males” (Table 1).

l 183: what is ESR1? please align with others

Ans: ESR1 is the gene name for ERα. Thus, we added the phrase “ERα mRNA” to the sentence and put “ESR1” into the following parenthesis (line 185, page 5).

l 195:  Esr1 knock down mice please align nomenclature

Ans: Because Esr1-knockdown muscle cells are generated by the Esr1-specific shRNA, we modified the sentence as below.

(old version) Experiments using Esr1-knockdown muscle cells implicates that impaired fatty acid oxidation results in the lipid accumulation in MERKO mice muscle [57].

(line 197, page 6 in revised version) Experiments using shRNA-mediated Esr1-knockdown muscle cells implicate that the impaired fatty acid oxidation, which may explain the lipid accumulation in MERKO mice muscle [57].

Reviewer 2 Report

Synopsis: In this manuscript under consideration (ijms-2157702), the authors reviewed the biological functions of estrogen, estrogen receptors, and estrogen-related receptors in modulating skeletal muscle physiology. This group has a substantial research foundation associated with the review content and thus can keep abreast of the recent development in this field. Overall, the manuscript is well-structured and well-written. There are only some minor questions that need to be improved prior to publication in the International Journal of Molecular Sciences.

1. Since the ERs and ERRs have shown great potential in the regulation of the development and regeneration of skeletal muscle, the authors can introduce some advances in therapeutic applications based on the ERs and ERRs, especially in the clinical trails and/or tissue engineering fields.

2. It would be more appealing for the reader if the authors use colorful figures instead of black-and-white graphs (e.g., Figures 1 and 2).  

3. There are some grammar and formatting mistakes throughout the manuscript. Please check carefully.

Author Response

Response to reviewers’ comments

Reviewer 2

Synopsis: In this manuscript under consideration (ijms-2157702), the authors reviewed the biological functions of estrogen, estrogen receptors, and estrogen-related receptors in modulating skeletal muscle physiology. This group has a substantial research foundation associated with the review content and thus can keep abreast of the recent development in this field. Overall, the manuscript is well-structured and well-written. There are only some minor questions that need to be improved prior to publication in the International Journal of Molecular Sciences.

Ans: We appreciate the reviewer' valuable comments and constructive suggestions, which help us improve the quality of the manuscript. We have carefully revised the manuscript according to these comments. Point-by-point responses are described below.

1. Since the ERs and ERRs have shown great potential in the regulation of the development and regeneration of skeletal muscle, the authors can introduce some advances in therapeutic applications based on the ERs and ERRs, especially in the clinical trails and/or tissue engineering fields.

Ans: As suggested by the Reviewer, we added the following description as new chapter 7 “Potential clinical implications of ER- and ERR-dependent muscle disease models and tissue engineering” in the revised manuscript (lines 417–437, page 11).

Current knowledge of ERs and ERRs on skeletal muscle will potentially provide a clue to the development of alternative therapeutic options, particularly in the context of tissue engineering. Studies using satellite cells isolated from OVX mice and satellite cell-specific ERb knockout mice show that the deficiency of estrogen signaling decreases the self-renewal and differentiation abilities in satellite cells [27,64]. In rats, estrogen increases the number of satellite cells after exercise [96,97] and stimulates muscle regeneration after injury in which ERb plays a primary role [98]. These findings imply that estrogen enhances the proliferation of muscle stem cells [99]. Interestingly, it is reported that estrogen can improve the production of functional myogenic differentiated cells from adipose-derived stem cells, which are used with a nano-scaffold to treat stress urinary incontinence [100]. In C2C12 cells, ERRα overexpression stimulates the myoblastic differentiation while ERRα inverse agonist XCT790 impairs myotube formation with fewer mitochondria and reduces sarcomeric assembly. These findings indicate that ERRα plays an important role in skeletal myocyte differentiation [101]. In addition, an ERRγ agonist enhances cardiac maturation with transverse tubule formation in cardiomyocytes derived from human induced pluripotent stem cells (hiPSCs) [102]. Thus, ERRγ signaling is also useful for muscle disease modeling and regenerative medicine. Further investigation will clarify the mechanisms of ERs and ERRs in muscle stem cells, which are critical for both fundamental tissue repair and regeneration.

2. It would be more appealing for the reader if the authors use colorful figures instead of black-and-white graphs (e.g., Figures 1 and 2).  

Ans: According to the reviewer’s suggestion, we colored the figures (Figs. 1 and 2).

3. There are some grammar and formatting mistakes throughout the manuscript. Please check carefully.

Ans: As suggested by the Reviewer, we carefully checked the manuscript throughout and corrected the mistakes/typos.

Reviewer 3 Report

The authors have reviewed and discussed a very actual and intriguing topic regarding the regulatory role of ERs and ERRs in skeletal muscle physiology, with highlight on mitochondria-related molecular mechanisms. This manuscript is clear and relevant to the field. The references cited are suitable and relevant, giving contemporary findings in this physiology. The statements and conclusions drawn are coherent and supported by the listed citations. The merit of the manuscript is good and follows a logical way.

Minor points:
English grammar and style still need to be double-checked. 

Author Response

Response to reviewers’ comments

Reviewer 3

The authors have reviewed and discussed a very actual and intriguing topic regarding the regulatory role of ERs and ERRs in skeletal muscle physiology, with highlight on mitochondria-related molecular mechanisms. This manuscript is clear and relevant to the field. The references cited are suitable and relevant, giving contemporary findings in this physiology. The statements and conclusions drawn are coherent and supported by the listed citations. The merit of the manuscript is good and follows a logical way.

Ans: We thank this reviewer for her/his valuable comments. As suggested by this reviewer, we respond to the comment.

Minor points:
English grammar and style still need to be double-checked. 

Ans: As suggested by the Reviewer, we carefully checked the manuscript throughout and corrected the mistakes/typos.